Stable isotope analyses of web-spinning spider assemblages along a headwater stream in Puerto Rico

Kelly Sean P. 1 spkelly.84@gmail.com
Cuevas Elvira 1
Ramírez Alonso 2
1 Department of Biology, University of Puerto Rico-Rio Piedras , San Juan, PR , USA
2 Department of Environmental Science, University of Puerto Rico-Rio Piedras , San Juan, PR , USA
Higley Leon
Electronic publication date: 2015 Oct 15
Publication date: 2015
Volume: 3
Electronic Location ID: e1324
Received 2015 Feb 6; Accepted 2015 Sep 24
Copyright: © 2015 Kelly et al.
Copyright year: 2015
Copyright holder: Kelly et al.
License: This is an open access article distributed under the terms of the Creative Commons Attribution License, which permits unrestricted use, distribution, reproduction and adaptation in any medium and for any purpose provided that it is properly attributed. For attribution, the original author(s), title, publication source (PeerJ) and either DOI or URL of the article must be cited.
License URL: https://creativecommons.org/licenses/by/4.0/

Keywords: El Yunque National Forest, Reciprocal subsidies, Aquatic insects, δ13C, Riparian zone, δ15N

Funding: Puerto Rico-Louis Stokes Alliance for Minority Participation NSF-HRD-1139888 Puerto Rico Center for Environmental Neuroscience NSF CREST HRD-1137725 Funding for stable isotope analyses was provided by the Puerto Rico-Louis Stokes Alliance for Minority Participation, (grant # NSF-HRD-1139888). Additional financial support during the writing process was provided by the Puerto Rico Center for Environmental Neuroscience (grant # NSF CREST HRD-1137725). The funders had no role in study design, data collection and analysis, decision to publish, or preparation of the manuscript.

==============================
Web-spinning spiders that inhabit stream channels are considered specialists of aquatic ecosystems and are major consumers of emerging aquatic insects, while other spider taxa are more commonly found in riparian forests and as a result may consume more terrestrial insects. To determine if there was a difference in spider taxa abundance between riverine web-spinning spider assemblages within the stream channel and the assemblages 10 m into the riparian forest, we compared abundances for all web-spinning spiders along a headwater stream in El Yunque National Forest in northeast Puerto Rico. By using a nonmetric dimensional scaling (NMDS) abundance analysis we were able to see a clear separation of the two spider assemblages. The second objective of the study was to determine if aquatic insects contributed more to the diet of the spider assemblages closest to the stream channel and therefore stable isotope analyses of δ15N and δ13C for web-spinning spiders along with their possible prey were utilized. The results of the Bayesian mixing model (SIAR) however showed little difference in the diets of riverine (0 m), riparian (10 m) and upland (25 m) spiders. We found that aquatic insects made up ∼50% of the diet for web-spinning spiders collected at 0 m, 10 m, and 25 m from the stream. This study highlights the importance of aquatic insects as a food source for web-spinning spiders despite the taxonomic differences in assemblages at different distances from the stream.

Introduction

Riparian zones have been identified as areas of high importance for maintaining biodiversity in aquatic and terrestrial habitats along with being an important interface for the exchange of resources, resulting in an ecosystem with unique environmental dynamics (Naiman & Decamps, 1997; Naiman, Decamps & Pollock, 1993; Nakano & Murakami, 2001). The importance of terrestrial subsidies (i.e., resources that are transported across ecosystem boundaries) as an energy source in the food webs of headwater streams has long been recognized (Vannote et al., 1980), but only more recently has it become evident that aquatic subsidies can be equally important in terrestrial food webs (Kato et al., 2003; Nakano & Murakami, 2001; Polis, Anderson & Holt, 1997; Sanzone, 2001; Sanzone et al., 2003). Emerging aquatic insects have been shown to be an important food source for a variety of terrestrial predators (Nakano & Murakami, 2001; Polis, Anderson & Holt, 1997) and the abundance of aquatic insects can affect the distribution of generalist predators such as insectivorous bats, reptiles, birds and spiders (Chan et al., 2008; Fukui et al., 2006; Iwata, Nakano & Murakami, 2003; Kato et al., 2003; Marczak & Richardson, 2007; Sabo & Power, 2002).

Web-spinning spiders are a particularly good model organism for studying the exchange of subsidies across riparian ecotones due to the fact that they are major consumers of emerging aquatic insects, and some taxa of web-spinning spiders have been associated exclusively with fresh water ecosystems (e.g., Tetragnatha (Tetragnathidae) and Wendilgarda (Theridiosomatidae) (Coddington, 1986; Eberhard-Crabtree, 1989; Gillespie, 1987). The distribution of these spiders has been correlated with aquatic insect abundances and for this reason these taxa of spiders are disproportionally more abundant within the first few meters from the stream channel where emerging insects tend to aggregate (Muehlbauer et al., 2014). The genus Tetragnatha has a worldwide distribution and can be found on all continents (except Antarctica) (Aiken & Coyle, 2000). Juvenile and female Tetragnatha typically construct relatively large, horizontal orb-webs directly above the surface of lentic and lotic bodies of freshwater (Gillespie, 1987). Wendilgarda is another genus of spider known to be associated with freshwater ecosystems; however, they are quite different from Tetragnatha in the sense that they are only found in tropical regions and the majority of taxa build a very reduced web structure that consists of one or two structural silk lines attached to rocks or vegetation along the stream with additional lines being attached to the water surface to snag drifting insects (Eberhard-Crabtree, 1989; Eberhard, 2001). Along with these aquatic specialists there has also been evidence that a variety of other taxa of web-spinning spiders (Araneidae, Lyniphiidae and Theridiidae) are also more abundant along streams where there are greater densities of aquatic insects (Marczak & Richardson, 2007).

Originally food web studies were generally conducted using observations in the field and gut content analyses, but recently the use of stable isotopes has become a preferred method for several reasons. One benefit of using stable isotopes is that gut content analyses are not viable methods for some organisms due to their feeding habits (e.g., spiders who feed on liquefied tissue) (Foelix, 2011). Another advantage of stable isotopes is that it is able to infer relatively long term feeding habits due to the bioaccumulation of δ15N and δ13C into the tissue of the consumer. A third advantage is that naturally occurring stable isotopes have been shown to be effective at identifying the contribution of different prey items in the diets of consumers through the use of mixing models (Parnell & Jackson, 2013; Peterson & Fry, 1987; Phillips & Gregg, 2003). This final aspect of stable isotope analyses is especially useful in aquatic and riparian food webs when determining the importance of subsidies that cross ecosystem boundaries, such as leaf litter falling into streams or emerging aquatic insects becoming food for terrestrial predators (Akamatsu, Toda & Okino, 2004; Burdon & Harding, 2008; Davis, Rosemond & Small, 2011; Hicks, 1997; Lau, Leung & Dudgeon, 2009; Sanzone, 2001; Sanzone et al., 2003; Walters, Fritz & Phillips, 2007).

This study had two main objectives. The first was to determine if there were differences in the composition of taxa in the assemblages of web-spinning spiders that were found over the stream channel compared to those 10 m from the stream. Because some spiders are specialists of aquatic ecosystems we predicted that these taxa would be in far greater abundance over the stream channel. The second objective was to determine if there were differences in the diets of these two assemblages using stable isotopic analyses. Most emerging aquatic insects remain very close to the stream channel and their abundance can drop exponentially only a few meters into the riparian area (Muehlbauer et al., 2014), so we predicted that the assemblage of web-spinning spiders in the stream channel would have a diet that reflects a greater dependence on aquatic insects while the assemblage in the riparian area would be feeding on a greater number of terrestrial insects.

Material and Methods

Study area

This study was conducted along the small headwater stream Quebrada Prieta within El Yunque National Forest in northeastern Puerto Rico at latitude 18°18′N and longitude 65°47′W (Masteller, 1993). The stream begins at around 600 m above sea level and runs into the Quebrada Sonadora at around 310 m above sea level with an average slope of 20% (Masteller, 1993). The stream ranges from 2–4 m in width and is mainly composed of large boulders and cobble with intermittent small pools with finer sediments of sand and silt. In 2012 total rainfall was 397.7 cm and the mean temperature was 23.93 (± 2.94) °C (Luquillo-LTER). The stream is surrounded by a mainly closed canopy of tabonuco (Dacryodes excels Vahl) forest which is the dominant tree species in the Luquillo Mountains up to 600 m in elevation (Masteller, 1993). Other common plant species include bullwood (Sloanea berteriana Choisy) and palms (Prestoea montana Graham and Nicolson)(Masteller, 1993).

The macroinvertebrate community of Quebrada Prieta is diverse and is composed of a variety of aquatic insects with the most abundant being trichopterans and ephemeropterans (Masteller, 1993). During a 52 week sampling period from 1990 to 1991, the total number of specimens collected in an emergence trap above Quebrada Prieta found 35% to be ephemeropterans, 24% trichopterans, 21% chironomids and 20% other dipterans (Masteller & Buzby, 1993a). In this study, emergence patterns varied somewhat between taxa, although as with most tropical streams, all taxa were present throughout the year (Ferrington, Buzby & Masteller, 1993; Flint & Masteller, 1993; Masteller & Buzby, 1993b; Pescador, Masteller & Buzby, 1993; Wagner & Masteller, 1993). For all taxa, the abundances were generally lowest during the summer months and highest during late fall to early spring (Ferrington, Buzby & Masteller, 1993; Flint & Masteller, 1993; Masteller & Buzby, 1993b; Pescador, Masteller & Buzby, 1993; Wagner & Masteller, 1993). For an example, ephemeropterans, trichopterans and chironomids were found to be the most abundant during November, March and January, respectively (Ferrington, Buzby & Masteller, 1993; Flint & Masteller, 1993; Pescador, Masteller & Buzby, 1993). All three taxa were found to be the least abundant during June and July (Ferrington, Buzby & Masteller, 1993; Flint & Masteller, 1993; Pescador, Masteller & Buzby, 1993).

Web-spinning spider assemblages

A 100 m reach of Quebrada Prieta was selected and then divided into four 25 m subsections. Field work for this portion of the project was conducted from April to August 2012, with at least one week between sampling dates to minimize the possibility of impacting the study area. A 3 m × 3 m riverine quadrat was selected within the stream channel in the first 25 m section of stream, measuring 3 m from the stream edge into the stream channel. Each quadrat was selected to contain a random mixture of available substrates, such as boulders, vegetation and deadwood which may affect web-spinning spider distribution. All web-spinning spiders within the quadrat up to 2.5 m in height were hand collected and preserved in 70% ethanol for later identification. This process was then repeated for a riparian site 10 m laterally from the stream edge into the riparian forest from the riverine sampling site. The same sampling procedure was then repeated for the next 25 m section of the stream. On the following sampling date we would sample the remaining two 25 m sections of the stream not sampled during the previous visit.

Each sampling date consisted of two riverine and two riparian quadrats. We conducted both diurnal and nocturnal sampling because some taxa of web-spinning spiders (e.g., Tetragnatha and Chrysometa) are more active at night and rarely build webs during the day. Nocturnal sampling on average was conducted from 19:00–23:00, while diurnal sampling on average was from 10:00–14:00. Sampling was only conducted during favorable weather conditions, because spider webs are many times easily destroyed by wind and rain (Foelix, 2011). Diurnal and nocturnal samplings were combined and therefore a total of eight riverine and eight riparian quadrats were analyzed for differences in web-spinning spider assemblages.

A Nonmetric Dimensional Scaling (NMDS) analysis along with a post-hoc Analysis of Similarity (ANOSIM) were used to determine if there were differences in taxa composition of the two web-spinning spider assemblages. A secondary post-hoc analysis, Similarity Percentages (SIMPER), was used to determine which particular taxa of spiders were causing a difference in the composition of the two assemblages. All of these analyses were conducted with the statistical program PAST (Hammer, Harper & Ryan, 2001).

Stable isotopes

To verify that the spider assemblages and their prey had stable isotope signatures that fell within realistic ranges of the basal C sources we sampled the three principal energy sources for aquatic and terrestrial arthropods. The three C sources sampled were stream leaf litter, periphyton and terrestrial vegetation. Stream leaf litter was collected at random throughout the 100 m stream transect and was gently rinsed to remove any macroinvertebrates. Periphyton was also sampled randomly by collecting rocks from the stream, gently rinsing them to remove any macroinvertebrates, and then scrubbing them with a small wire brush. The resultant slurry was then collected into glass vials to be dried later. For terrestrial vegetation samples, green leaves were collected at random from C3 plants within the riparian forest.

Possible insect prey of the spider assemblages were collected for isotope analysis using two methods. Flying insects were collected using a passive sampling method with three Malaise traps that were placed within the stream channel for approximately four hours during the diurnal and nocturnal spider sampling. Traps were placed within three different sections of the stream reach (0–25 m, 25–75 m and 75–100 m) during each sampling period. This was done in order to have a representation of the available prey of web-spinning spiders flying along the stream channel. Aquatic insect larvae were collected using hand nets throughout the 100 m stream reach. Sampling was conducted in pools, riffles, and cascades to ensure that all major microhabitats were sampled. The larval stages of Ephemeroptera, Trichoptera and Chironomidae were used for isotopic analysis because they no longer feed as adults and thus their isotopic signature is fixed during the aquatic larval stage.

To compare the δ15N and δ13C stable isotope signals for the different spider assemblages, individuals were collected from a riverine transect within the stream channel, from a riparian transect 10 m parallel from the stream edge and from an upland transect 25 m parallel from the stream edge. In each transect, web-spinning spiders were collected from the four most abundant families: Tetragnathidae, Theridiosomatidae, Pholcidae and Uloboridae. Spiders were collected and maintained live in small containers for a day, to allow for the digestion of prey that may have been recently consumed to reduce the influence of the isotopic signal from what they were consuming.

Specimens were frozen (−20 °C) for a minimum of 24 h, then placed in a drying oven for a minimum of 48 h (70 °C) and finally ground to a fine powder for isotopic analysis. Insects were identified to family (except for Lepidoptera identified to order) and spiders were identified to genus (except for Wendilgarda clara Keyserling 1886, identified to species). Composite taxa samples of a minimum of four individuals for spiders 1 ± 0.05 mg of animal tissue and 5 ± 0.05 mg of plant tissue was measured for the natural abundances of 15N and 13C using ratio mass spectrometry at the Miami Stable Isotope Ecology Lab at the University of Miami in Florida. Natural abundances of stable isotopes for δ13C and δ15N were calculated as: δ13C or δ15N=Rsample/Rstandard−1×1,000

where, Rsample = 13C:12C or 15N:14N ratio in the sample and Rstandard = 13C/12C ratio in Pee Dee Belemnite for δ13C and Rstandard = 15N/14N ratio in the atmosphere for δ15N (Peterson & Fry, 1987).

The stable isotopes 15N and 13C of insects were analyzed as composite samples with aquatic insect taxa compiled first by family into one of five functional feeding groups: collector-gatherers (n = 1), filterers (n = 2), predators (n = 3), scrapers (n = 2) and shredders (n = 2) (Ramirez & Gutierrez-Fonseca, 2014). Terrestrial insects were grouped as either herbivorous (n = 3) or predacious (n = 2). Terrestrial dipterans were only identified to order, and due to their varied feeding behaviors they were not placed into a particular feeding group. The isotopic values for the five aquatic functional groups were then combined into a single aquatic insect group and the values for the three terrestrial insect groups (predators, herbivores and dipterans) were combined as well. Spider taxa were identified to genus and were grouped as either having been collected in riverine (n = 7), riparian (n = 5) or upland (n = 5) transects. Mean averages of δ13C and δ15N for each group (aquatic insects, terrestrial insects, riverine spiders, riparian spiders and upland spiders) were used in subsequent biplots and dietary analyses.

Dietary analyses were conducted utilizing Bayesian mixing models in the SIAR package version 4.2 with Stable Isotope Analysis in R (SIAR) (Parnell & Jackson, 2013) for R version 3.0.3 (R Core Team, 2012). Consumers were the three spider groups (riverine, riparian and upland) and the sources were the two insect groups (aquatic and terrestrial). Fractionation factors between consumers and sources (δ13C: 0.08 ± SD 1.90 and δ15N: 2.75 ± SD 2.20) were adopted from the work by Yuen & Dudgeon (in press) in which they had reviewed fractionation values for arthropod consumers from a previous comprehensive study (Caut, Angulo & Courchamp, 2009). The proportion of aquatic insects in the diets of the three spider groups were determined from the SIAR package that provides 5, 25, 75 and 95% credibility intervals from the Bayesian mixing models.

Results

Web-spinning spider assemblages

Four diurnal and four nocturnal samplings were conducted for both riverine and riparian habitats. Five families of web-spinning spiders (Araneidae, Pholcidae, Tetragnathidae, Theridiosomatidae and Uloboridae) were collected in varying abundances from riverine and riparian quadrats (Table 1). The least abundant family was Araneidae with only two individuals collected, while the family Theridiosomatidae was the most abundant with 199 individuals collected from two taxa, Theridiosoma sp. and Wendilgarda clara (Keyserling)  (Table 1). The second most abundant family was Tetragnathidae with 146 individuals collected from three genera, Chrysometa, Leucauge and Tetragnatha (Table 1). Uloboridae was the third most abundant family with 32 individuals collected from the Miagrammopes genus (Table 1). Pholcidae was the second to least abundant family with 28 individuals collected from the Modisimus genus (Table 1). There were 265 spiders collected from the riverine habitat while in the riparian habitat 142 spiders were collected.

Table 1 Web-spinning spider abundance.

Average number of individuals for each spider taxa collected from the eight riverine and riparian quadrats.

Family	Genus	Riverine spiders (mean ± STDEV)	Riparian spiders (mean ± STDEV)	
Araneidae		0.1 ± 0.4	0.1 ± 0.4	
Pholcidae				
	Modisimus	2.5 ± 1.7	1.0 ± 0.9	
Tetragnathidae				
	Chrysometa	2.6 ± 4.7	0.9 ± 1.5	
	Leucauge	6.3 ± 6.3	7.8 ± 7.5	
	Tetragnatha	0.8 ± 0.9	0.0 ± 0.0	
Theridiosomatidae				
	Theridiosoma	0.6 ± 0.7	2.8 ± 2.1	
	Wendilgarda a	19.4 ± 9.0	2.1 ± 2.2	
Uloboridae				
	Miagrammopes	0.9 ± 0.6	3.1 ± 3.1	
Notes.

a Identified to species, Wendilgarda clara (Keyserling, 1886).

A NMDS analysis of the two web-spinning spider assemblages shows a clear spatial separation of the eight riparian and eight riverine groups (Fig. 1). This was statistically verified with the post-hoc test ANOSIM, which showed a significant difference in the degree of separation between the two assemblages (Bonferroni-corrected, p < 0.002, R = 0.722) (Fig. 1). An additional post-hoc analysis, SIMPER, found that around 48% of the dissimilarity between the assemblages was attributed to the abundance of Wendilgarda clara.

Figure 1 NMDS analysis.

NMDS of web-spinning spider abundances for each sampling date. Riparian (Rip) quadrats and riverine (Riv) quadrats. Bray-Curtis 95% ellipses. ANOSIM Bonferroni-corrected p = 0.002, R = 0.7224.

Basal carbon sources and prey taxa

Stable isotope analyses of the basal C sources showed a difference of δ13C in terrestrial vegetation, periphyton and stream leaf litter. Terrestrial vegetation (−34.90‰) was more depleted in δ13C than aquatic periphyton (−32.40‰) and stream leaf litter (−25.50‰) (Table 2). δ15N values were very similar for C3 vegetation (−1.30‰) and periphyton (−0.80‰) while stream leaf litter had the highest δ15N value (0.80‰). Despite these differences in δ13C, there was no clear separation between aquatic and terrestrial C signatures.

Table 2 Stable isotope values.

Carbon (δ13C) and nitrogen (δ15N) stable isotope values of all samples used in subsequent analyses.

	Functional feeding group (FFG)	Order	Family	Genus	δ 13 C	δ 15 N	
Stream leaf litter					−25.50	0.80	
Forest vegetation					−34.90	−1.30	
Stream periphyton					−32.40	−0.80	
Aquatic insects	Collector-gatherer	Diptera	Chironomidae		−26.63	2.63	
	Filterer	Diptera	Simuliidae		−27.76	2.62	
	Scraper	Ephemeroptera	Leptophlebidae		−28.15	2.59	
	Predator	Odonata	Coenagrionidae		−27.42	4.98	
	Shredder	Trichoptera	Calamoceratidae		−28.55	0.78	
	Scraper	Trichoptera	Helicopsychidae		−34.88	1.96	
	Predator	Trichoptera	Hydrobiosidae		−26.51	5.09	
	Filterer	Trichoptera	Hydropsychidae		−29.41	3.69	
Terrestrial insects	Predator	Coleoptera	Lampyridae		−25.30	6.31	
		Diptera			−26.75	4.32	
	Herbivore	Hemiptera	Cicadoidea		−28.69	−0.55	
	Predator	Hymenoptera	Evaniidae		−26.82	3.83	
	Herbivore	Lepidoptera			−27.40	1.94	
Riverine spiders		Aranea	Pholcidae	Modisimus	−26.65	3.40	
		Aranea	Tetragnathidae	Chrysometa	−26.72	5.19	
		Aranea	Tetragnathidae	Leucauge	−26.66	4.53	
		Aranea	Tetragnathidae	Tetragnatha	−27.54	3.87	
		Aranea	Theridiosomatidae	Theridiosoma	−27.34	3.84	
		Aranea	Theridiosomatidae	Wendilgarda a	−27.24	4.24	
		Aranea	Uloboridae	Miagrammopes	−27.35	2.90	
Riparian spiders		Aranea	Pholcidae	Modisimus	−28.49	3.44	
		Aranea	Tetragnathidae	Chrysometa	−27.40	4.51	
		Aranea	Tetragnathidae	Leucauge	−27.25	4.76	
		Aranea	Theridiosomatidae	Theridiosoma	−27.74	3.55	
		Aranea	Uloboridae	Miagrammopes	−27.24	2.54	
Upland spiders		Aranea	Pholcidae	Modisimus	−27.50	3.80	
		Aranea	Tetragnathidae	Chrysometa	−26.90	5.00	
		Aranea	Tetragnathidae	Leucauge	−27.30	3.80	
		Aranea	Theridiosomatidae	Theridiosoma	−26.50	1.90	
		Aranea	Uloboridae	Miagrammopes	−33.30	0.10	
Notes.

a Identified to species, Wendilgarda clara (Keyserling).

δ13C and δ15N values for individual insect groups varied among taxa. The family Helicopsychidae (Trichoptera) was the most depleted in δ13C (−34.88‰), while the family Lampyridae (Coleoptera) was the most enriched in δ13C (−25.30‰) (Table 2). The family Cicadoidea (Hemiptera) had the lowest δ15N value (−0.55‰), while Lampyridae, a terrestrial predator, was not only the most enriched in δ13C but was also the most enriched in δ15N (6.31‰) (Table 2). There was also a large amount of variation seen in the δ13C and δ15N values when insects were analyzed according to their functional feeding groups. The terrestrial predator group of insects was the most enriched in δ13C (−26.06 ± SD1.75‰), followed by collector-gatherers (−26.63‰), aquatic predators (−27.26 ± SD0.68‰), terrestrial herbivores (−28.05 ± SD0.91‰), shredders (−28.55‰), filterers (−28.59 ± SD1.17‰) and scrapers (−31.52 ± SD4.75‰). Terrestrial predators (5.07 ± SD1.75‰) were the most enriched in δ15N, followed by aquatic predators (4.35 ± SD1.20‰), filterers (3.16 ± SD0.76‰), collector-gatherers (2.63‰), scrapers (2.27 ± SD0.45‰), shredders (0.78‰) and terrestrial herbivores (0.69 ± SD1.76‰). When insect taxa were grouped together and analyzed as either terrestrial (n = 5) or aquatic (n = 10), no significant difference was found in δ13C and δ15N values between the two groups (Fig. 2). Although there was no significant difference in δ13C, terrestrial insects (−26.99 ± SD1.22‰) were overall more enriched than aquatic insects (−28.66 ± SD2.69‰). Terrestrial insects (3.17 ± SD2.60‰) were also more enriched in δ15N although they also showed greater variation than aquatic insects (3.04 ± SD1.47‰).

Figure 2 Biplot of stable isotope values.

Biplot of carbon (δ13C) and nitrogen (δ15N) stable isotope values of basal resources and consumers. Points are mean values with error bars representing standard deviation. Basal resources are single samples and therefore without error bars. Spiders (squares), insects (circles), basal resources (triangles).

Web-spinning spiders

Stable isotope analyses of the individual spider taxa showed less variation in δ13C and δ15N values than was seen in the insect taxa. The genus Miagrammopes (Uloboridae) along the upland transect was the most depleted in δ13C (−33.30‰), while Theridiosoma] (Theridiosomatidae) also from the upland transect was the most enriched in δ13C (−26.50‰) (Table 2). The genus Chrysometa (Tetragnathidae) from the riverine transect had the highest δ15N value (5.19‰), while upland Miagrammopes (Uloboridae) had the lowest δ15N value (0.10‰) (Table 2). There were no significant differences in δ13C and δ15N values between the three spider groups. The group the most enriched in δ13C were riverine spiders (−27.07 ± SD0.38‰), followed by riparian spiders (−27.62 ± SD0.52‰) and finally upland spiders (−28.30 ± SD2.82‰) (Fig. 2). Similarly, the group with the highest δ15N values were riverine spiders (4.00 ± SD0.75‰), followed by riparian spiders (3.76 ± SD0.89‰) and finally upland spiders (2.92 ± SD1.93‰) (Fig. 2). The greatest amount of variation in both δ13C and δ15N values was seen in upland spiders (Fig. 2).

Bayesian mixing model analyses

Bayesian mixing models determined that the proportion of aquatic insects in the diets of the three spider groups was relatively similar. However, the proportion of aquatic insects in the diets of the spiders increased slightly in the groups further away from the stream channel (Fig. 3). The analyses revealed that riverine spiders had the least amount of aquatic insects in their diet (45–47%) although there was considerable variation (12–71%) (Fig. 3). In the riparian spider group, aquatic insects made up a slightly greater proportion (47–49%), but again a great deal of variation was seen (10–80%) (Fig. 3). The proportion of aquatic insects was greatest in the upland spiders (50–53%) although this group also had the greatest amount of variation (10–98%) (Fig. 3).

Figure 3 Bayesian mixing model dietary analysis.

Boxplots with 5, 25, 75 and 95% credibility intervals representing the proportion of aquatic insects in the diets of riverine, riparian and upland spiders.

Discussion

The influence of emerging aquatic insects has been shown to affect web-spinning spider distributions in riparian areas, especially within the first 10 m from the stream edge (Collier, Bury & Gibbs, 2002; Kato et al., 2003; Kato, Iwata & Wada, 2004; Sanzone et al., 2003). Most emerging aquatic insects follow a negative power function abundance curve and over 50% of their “signature” has been found to be within only 1.5 m from the stream, although some variation has been found depending on the taxa of aquatic insect (Muehlbauer et al., 2014). We established our working hypotheses based on the strong link between web-spinning spiders and emerging aquatic insects and the fact that the majority of the insects congregate within only a few meters of the stream edge. First we proposed that there would be a different assemblage of web-spinning spiders, due to the presence of aquatic specialists (Tetragnatha and Wendilgarda), within the stream corridor compared to 10 m into the riparian forest. We then proposed that because the majority of aquatic insects congregate within only a few meters of the stream, that the riverine spider assemblage in the stream corridor would be consuming more aquatic insects than riparian and upland spiders. We found that there was indeed a significant difference between the riverine and riparian assemblages and that around 48% of the dissimilarity between the assemblages was attributed to the abundance of Wendilgarda, a specialist of aquatic habitats. The results did not entirely support our second hypothesis. The analyses of stable isotopes showed no clear separation between the δ13C signature for aquatic and terrestrial prey due to the fact that the aquatic food web was driven by leaf litter inputs from the terrestrial vegetation that resulted in similar δ13C ranges for both terrestrial and aquatic primary consumers. As a result, the biplot of δ13C and δ15N showed significant overlapping of the three spider groups along with the aquatic and terrestrial insects. This resulted in the inability to visually separate the consumer groups or their prey. The Bayesian SIAR dietary analysis showed that the upland group of spiders relied the most heavily on aquatic insects (50–53%) although only slightly more so than riverine (45–47%) and riparian (47–49%) spiders (Fig. 3). Overall, aquatic insects were found to be an important food source for web-spinning spiders even up to 25 m from the stream channel.

The difference in assemblage composition between the stream channel and the riparian forest was found to be driven mainly by an aquatic specialist, Wendilgarda, which snare their prey directly from the water surface (Coddington, 1986; Eberhard-Crabtree, 1989). Tetragnatha, another aquatic specialist (Aiken & Coyle, 2000; Alvarez-padilla & Hormiga, 2011; Gillespie, 1987), was also only found only in riverine quadrats however there were too few individuals to have any statistical significance. Studies of riparian spider assemblages in other parts of the world have found similar shifts in taxa composition, in which the abundance of some spiders was directly related to the distance from the stream edge and that significant differences could be found within only 10 m into the riparian zone (Sanzone, 2001; Sanzone et al., 2003). However, most studies have only been conducted in temperate regions and so far few studies that have investigated whether this distribution of spider taxa also occurs along tropical streams. Some of the proposed biotic and abiotic factors that could explain the shift in spider distributions range from differences in vegetative complexity and structure (Chan, Zhang & Dudgeon, 2009) to changes in humidity and temperature, but the most common factor associated with the distribution of web spinning spiders has been associated with the abundance of aquatic insects (Kato et al., 2003; Kato, Iwata & Wada, 2004; Sanzone, 2001; Sanzone et al., 2003).

Basal carbon sources (stream leaf litter, periphyton and C3 vegetation), prey items (terrestrial and aquatic insects) and web spinning spiders (riverine, riparian and upland) (Table 2) were all found to have isotopic signals within the range of reported values from other studies (Fry, 1991; Kato, Iwata & Wada, 2004; Lau, Leung & Dudgeon, 2009; March & Pringle, 2003; Ometto et al., 2006; Trudeau, 2003). Terrestrial vegetation was the most depleted in δ13C and δ15N, stream leaf litter was the most enriched in δ13C and δ15N and periphyton was the intermediate of the two (Table 2). The difference between the stable isotopic signals of the stream leaf litter and terrestrial vegetation could be a result of the stream leaf litter having been derived from other vegetation found upstream that were not necessarily present long the section of stream that was sampled during the study. The differences in the type of terrestrial vegetation found upstream could account for some of the difference in isotopic values. Allochthonous and autochthonous C sources in riparian food webs can vary considerably in their δ13C signature (±10‰) depending on several factors such as plant taxa, water velocity, and canopy cover (Lau, Leung & Dudgeon, 2009; March & Pringle, 2003; Ometto et al., 2006; Trudeau, 2003). Basal carbon sources were utilized in determining a reasonable range in which subsequent consumers should be found.

Isotopic values of insect taxa were all found to be within the range of basal C sources; however there was no clear separation in the isotopic signals between terrestrial and aquatic insects (Fig. 2). Terrestrial predators and herbivores showed little variation in their δ13C signal, −28.05 ± 0.91‰ and −26.06 ± 1.08‰ respectively. The enriched δ13C signal in the predators is most likely associated with bioaccumulation more so than a change in C sources. Of all the insect groups, terrestrial herbivores and predators had the lowest and highest δ15N values, respectively, similar to what was reported in a study done by Kato, Iwata & Wada (2004) in Japan where they also found a difference of around 4‰ between terrestrial herbivores and predators (Kato, Iwata & Wada, 2004).

The δ13C signature for the aquatic insect groups, as mentioned earlier, was not statistically different from the terrestrial insects and most of the functional feeding groups had overlapping values with terrestrial herbivores emphasizing the importance of leaf litter inputs in the aquatic food web. Overall, aquatic insects were more depleted in δ13C and δ15N than terrestrial insects, similarly to what has been found in other studies (Kato, Iwata & Wada, 2004). Scrapers were found to be the most depleted in δ13C and this group also showed the greatest range in their δ13C signature (−1.52 ± 4.75). This variation is most likely the result of the two taxa that were collected for this functional group. Helicopsychidae were severely depleted in δ13C due to them being obligate scrapers, feeding on C sources depleted in δ13C such as periphyton and possibly other more depleted C sources that were not sampled in this study (e.g., aquatic moss). Leptophlebiidae are considered to be more generalists and at times may feed as collector-gatherers, despite the families overall classification as scrapers (Ramirez & Gutierrez-Fonseca, 2014). Similar isotopic values were found for Leptophlebiidae in a nearby stream (δ13C: −24.25 ± 0.72‰ and δ15N: 2.51 ± 0.20‰) (March & Pringle, 2003). The small change in δ13C could have been a result of the stream in the study by March & Pringle (2003) having a more open canopy and therefore a possible greater presence of algae. The aquatic insect groups had δ15N signatures that fell within the two terrestrial extremes with aquatic predators (4.35 ± 1.20‰) and shredders (0.78‰) having respectively the highest and lowest δ15N signatures. Collector-gatherers, scrapers and filterers were found to be intermediary with relatively little variation in their δ15N values (1.96–3.69‰).

The three spider groups showed only slight differences in their δ13C and δ15N signatures with upland spiders being the most depleted in both instances. A study conducted in Japan similarly found only minimal changes in the δ13C and δ15N values between riparian and upland web spinning spiders of the same taxa (Kato, Iwata & Wada, 2004). However, in our study we did not analyze individual taxa and included only web-spinning spiders. This may explain some of the similarity between riverine, riparian and upland groups. Other studies have found that differences in stable isotopes values can be associated with different hunting strategies (i.e., sit and wait, wandering or web building) (Collier, Bury & Gibbs, 2002; Kato, Iwata & Wada, 2004; Sanzone et al., 2003; Yuen & Dudgeon, in press).

The Bayesian analysis in SIAR found that upland spiders relied the most upon aquatic insects although there were only slight differences among the three spider groups. Although the vast majority of aquatic insect biomass is concentrated within the first few meters or so from the stream edge, some taxa are known to disperse laterally up to hundreds of meters from the stream (Muehlbauer et al., 2014). For an example, even at around 13.3 m from the stream edge an estimated 50% of the abundance of chironomids would still be present (Muehlbauer et al., 2014). Around 10% of the abundance for ephemeropterans and trichopterans was estimated to be found still even 160 m and 650 m respectively from the stream edge (Muehlbauer et al., 2014). We found that aquatic insects made up around 50% of the spiders’ diet which was slightly less than what has been reported in some other studies which have found that aquatic insects can make up ∼70–90% of the diet of riparian spiders (Akamatsu, Toda & Okino, 2004; Sanzone et al., 2003). The dominant riparian taxa in those studies however were species of Tetragantha, which have been found to be specialists in trapping emerging aquatic insects. Along our site there were extremely few Tetragnatha and were therefore not included in our isotopic analyses. Our results were more similar to those reported for other tropical (Yuen & Dudgeon, in press) and sub-tropical (Collier, Bury & Gibbs, 2002) sites. In Hong Kong,Yuen & Dudgeon (in press) found that riparian web-building spiders had a mean dependence of ∼36–55% on aquatic insects. In New Zealand, Collier, Bury & Gibbs (2002) found that the mean contribution of aquatic insects to all riparian spider taxa was ∼58%. Emergence patterns of aquatic insects can vary greatly among and even within tropical, sub-tropical and temperate streams and this could have a large influence on the importance of stream subsidies to surrounding terrestrial predators. Some of the variability found among studies may be related to the type of isotopic mixing model that was applied (linear, algorithmic, or Bayesian), differences in which spider taxa were present, or differences between stream localities (tropical, subtropical and/or temperate).

Our study highlights the importance of riparian ecotones as areas that contain a unique biodiversity of web-spinning spider taxa that are specialists in aquatic habitats and are rarely found even after only a few meters from the water’s edge. Dietary analyses revealed that aquatic insects comprised ∼50% of the diet in riverine, riparian and upland spiders with only a slightly greater dependence on aquatic insects in the upland spider group. We found that isotopic signals between terrestrial and aquatic insects were not exclusively distinct and this can impact the effectiveness of isotopic mixing models, which has been shown to be a problem for other studies along forested headwater streams. Despite the overlapping of isotopic signals, the results of the dietary analysis were similar to other studies conducted along tropical streams. Our study provides further evidence for the importance of aquatic subsidies for terrestrial consumers even within upland areas from the stream.

Conclusion

The environment provided by the stream channel and that of the riparian forest clearly created two unique web-spinning spider assemblages, in which specialized taxa of aquatic ecosystems were shown to be the major difference between the two study areas. However, differences between these two habitats were potentially the result of structure and microenvironment, rather than prey resources.

Supplemental Information

Supplemental Information 1 Supplemental data

Click here for additional data file.

We thank Dr. Ingi Agnarsson for his assistance in spider identification, Pablo Gutiérrez for assistance in aquatic insect identification, Dr. Leonel Sternberg for his help in the analysis of stable isotopes, and Alexandra Salcedo and Adolfo Barragán for their help with fieldwork.

Additional Information and Declarations

Competing Interests

Author Contributions

The authors declare there are no competing interests.

Sean P. Kelly conceived and designed the experiments, performed the experiments, analyzed the data, wrote the paper, prepared figures and/or tables, reviewed drafts of the paper.

Elvira Cuevas and Alonso Ramírez analyzed the data, contributed reagents/materials/analysis tools, reviewed drafts of the paper.

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
