# Peer review of "Stable isotope analyses of web-spinning spider assemblages along a headwater stream in Puerto Rico"

_PeerJ, doi:10.7717/peerj.1324_

## Round 0.1 · original submission · Major Revisions

I have included a pdf version of your submission in which I made edits, and explained my suggested changes. Here, I mostly focused on grammatical or stylistic points. I think both reviewers offer valuable suggestions for improving the manuscript. As both reviewers had some reservations/suggestions regarding analysis, please pay particular attention in addressing these.

·

Basic reporting

Yes, it's OK, but Figure 2 was rather messy-looking on the version of the MS files available for review.

Experimental design

No particular concerns. Pity there was no seasonal component, as this makes it hard to asses the significance of the results within the context of any annual cycle.

Validity of the findings

I am a little concerned about the analysis of the stable-isotope data, and the model used to assess the relative contribution of potential foods to the predators. It is no longer 'state-of-the art'. My report (below) gives further details of this. Also I am worried that many more samples of aquatic insects were analysed than terrestrial counterparts. This could mean that the variety of terrestrial signals is underrepresented in the data set. A third thing is why didn't the authors sample spiders further inland?

Additional comments

Stable isotope analyses of web-spinning spider assemblages along a headwater stream in Puerto Rico

This is a very simple review for me to write, as I’ve worked on riparian spiders with my students, and I know well and greatly respect the work of one of the authors (Ramirez). In addition, my research group are tackling exactly the same problems as Kelly et al.: i.e. spiders in tropical stream riparia. We have used exactly the same methods (SIA mixing models) to assess the relative dependence of different spiders on aquatic prey, and looked at how this varies with distance from the stream. The work could hardly be more similar, with the primary difference being that my group are now working with ground-dwelling spiders (we previously looked at tetragnathids) whereas Kelly et al. are researching web-spinners.

With that background, I am a bit surprised that the authors did not cite some of our work on, for instance, the distribution of aquatic insects with increasing distance from stream margins or the role of stream channels as hotspots for web-building spiders (see list at end of this review). We also found evidence that stream channels were feeding sites for birds that made use of emerging aquatic insects.

At any rate, I think that the research question is interesting and the approach used is appropriate. The analyses seem fine and the presentation of the results is also adequate. The MS is generally well prepared. I have some concern about aspects of the sample base for the SIA and the mixing models used and the results (see below) but, assuming the authors can address this, I recommend acceptance, after some moderate to major revision. The authors might also dial back the rather grandiose statements about reciprocal subsidies at the end of the MS. They need a bit more data that they have in had now to warrant such claims.

Specific comments are set out below.

Pg. 3, lines 29 and 30: this reads as though the spiders lived underwater!

Bottom of pg 4: the reader only needs to know about the insects as they are the potential food sources for the spiders. Are there any data on the composition, magnitude and seasonality of emergence? This is the one major gap I see in this study. Only one study is mentioned that shows a decline in insects with distance from the stream; a bit more supporting evidence would be good (e.g. Chan et al. 2007). It is also a pity that the study did not go further inland than 10m. Another more landward transect would have been informative as aquatic insects do go further inland than 10 m..

P6 5: is it right that there were 4 sampling dates? This is implicit from the number of samples collected (4 x 4 at each distance), but the actual sampling dates or intervals are not given (just the overall duration, and mention of at least one-week separation). Is there any way to convince the reader that the sampling effort was adequate (is 4 quadrats on each occasion sufficient)? The numbers of spiders collected (from the SI) is not very large (perhaps this is inevitable given they are predators).

Line 16-17, pg. 6: are these distances the location of the traps along the study reach within the channel? The way the text is written could benefit from clarification.

Pg 7, line 18-22. Are the authors concerned that the SIA analyses of potential prey contained more taxa/samples that the terrestrial prey? This is obvious from the SI, but also it seems clear frpm the text that there were only 4 composite (FFG) samples of terrestrial prey analysed, yet 10 composite samples of aquatic prey. I worry that the authors have not sampled enough of potential terrestrial prey (Malaise traps could have been placed along the 10 m quadrats to do this) and thus the range of isotopic signatures of potential terrestrial prey has been underestimated relative to signatures of their aquatic counterparts. I think the authors need to address this point and consider how it might have affected their conclusions. It could have led to an underestimate of the importance of terrestrial insects to spider diets. Why didn't they sample more terrestrial insects?

Pg. 7, line 27: The mixing models used in this study (based on IsoSource) are now rather out of date: the state of the art models are Bayesian mixing models in SIAR package ver 4.2 (Parnell and Jackson 2013), and I would recommend that the authors check their results against these more up-to-date models. We have found these newer models to be far more informative that the older approach.
Parnell, A. and A. L. Jackson. 2013. siar: stable isotope analysis in R. R package version 4.2. URL http://CRAN.R-project.org/package=siar

In this connection, I would caution about assumptions on fractionation or N-enrichment values, which are typically assumed to be constant for predators (e.g. spiders) eating prey (e.g. insects). They are not constant, and the assumption about this fractionation can affect the outcomes of SIA studies. While there is probably not much the authors can do about this, it is a study limitation and maybe should be acknowledged as such. See Caut et al. (2009) for an important review of this matter.
Caut, S., E. Angulo, and F. Courchamp. 2009. Variation in discrimination factors (Δ15N and Δ13C): the effect of diet isotopic values and applications for diet reconstruction. Journal of Applied Ecology 46:443-453.

Page 8 and elsewhere: the Shannon-Wiener data add nothing and can be deleted. What is the reader supposed to conclude from the two values presented: 1.30 and 1.56? In my view, this is an overused and not especially informative diversity index. (BTW, using the exponent of the SWI can be much more informative if one has a lot of samples to compare.)

Bottom of page 8: how do the authors explain the large difference in C signature between terrestrial leave and aquatic leaf litter. The disparity is much greater than I have ever seen in similar studies, and the periphyton is much closer to terrestrial leaves than terrestrial leaves are to terrestrial leaves that had fallen in the stream and become litter. This is odd! And, presumably, because terrestrial leaves are like periphyton, and submerged litter is different, the authors conclude (top of page 9) that there was no clear difference between aquatic and terrestrial C signatures of basal foods nor in the signatures of the aquatic and terrestrial insects.

I would note here that we have found a similar lack of difference (or big overlap) in signatures of aquatic and terrestrial insects, but some terrestrial insects are very different from aquatic insects, and it depends which terrestrial insects you are looking at. This is why I was a bit concerned that they authors only have 4 SIA samples of terrestrial insects.

Pg 10, line 8-13: this is the ‘meat’ of the results, and it seems that the contribution of aquatic and terrestrial insects to spider diets is very similar irrespective if you look at the spiders along the stream or those in the forest. For instance the contribution of terrestrial herbivores to spiders along the stream is 0-65% and those in the forest is slightly lower 0-55%. Since the respective figures of terrestrial predators is 0-25% and 0-26% we can actually see that the spiders along the stream ate MORE terrestrial insects than those inland. Of course, the ranges of all of these values are very wide, so minor differences should not be overstated.

Fig 2: this could have been presented somewhat better (it’s an unattractive graphic), and I note the ordering of the panels does not match that stated in the legend. Rather than showing just the distribution of possible percentage contributions, can the authors tease out the most likely contribution for each food type, and then work out the overall contributions of terrestrial and aquatic food sources? The Bayesian model I mention above tends to give narrower % ranges (e.g. the solutions do not all include 0%).

Discussion: I wonder if the authors’ results would have changed if they had sampled litter-feeding terrestrial insects. They suggest that aquatic shredders were the most important food of spiders but the range of possible solutions (0-93% in one case) is very wide, and flying insects that feed on litter could have been captured by web-spinners.

Pg 12, lines 6-9: I cannot understand this sentence; please clarify or reword. Are the authors saying they did not make any real use of the basal-food-signature data? (This appears to be the case.)

Discussion, latter part: overstates the results a bit (esp. the last paragraph); yes there is some contribution of aquatic insects to the diet of spiders 10 m from the stream, but what about further inland? Why didn't the authors establish another transect further inland where there would have been many fewer aquatic insects? Also, the aquatic contribution is not that unexpected so close to the stream. What was the relative abundance of aquatic and terrestrial prey? How would it have changed with season? The evidence for reciprocal subsides is not as strong as the authors’ state and really needs a year-round investigation to make the relative importance of the water-to-land energy flow more clear.

The Conclusions section is repetitious of the last part of the Discussion and is not needed.

The authors may want to consult the following papers that are relevant to their study.
Chan, E.K.W., Zhang, Y. & Dudgeon, D. (2007). Contribution of adult aquatic insects to riparian prey availability along tropical forest streams. Marine & Freshwater Research 58: 725-732
Chan, E.K.W., Tung, Y.-T., Zhang, Y. & Dudgeon, D. (2008). Distribution patterns of birds and insect prey in a tropical riparian forest. Biotropica 40: 623-629
Chan, E.K.W., Zhang, Y. & Dudgeon, D. (2009). Substrate availability may be more important than aquatic insects in the distribution of riparian orb-web spiders in the seasonal tropics. Biotropica 41: 196-201

David Dudgeon (18 Feb, 2015)

Reviewer 2 ·

Basic reporting

The work needs some editing and the authors should consider new ways to analyze/ present the data.

Experimental design

Authors need to clarify their methods. The authors need to justify their analysis or re-analyze based on temporal differences.

Validity of the findings

Project is limited in scope but could add to the literature.

Additional comments

The manuscript, “Stable isotope analyses of web-spinning spider assemblages along a headwater stream in Puerto Rico” reports the findings of researchers who analyzed spiders living over or beside a small river or 10 meters distant from the river in the riparian forest. The results of the study were that spider assemblage differed (as would be anticipated) but prey base was very similar suggesting that emerging aquatic insects contributed substantially to spider diet even some distance from the stream.
Overall, the body of the paper is generally well-written, however the materials and methods need clarification. The applicability of the results to wider scales are also not clear as the authors sampled only one watershed in one year. At a minimum the authors should re-visit the discussion to highlight the limits of this data set.
I recommend that the paper be revised with particular attention paid to the comments below:
Major concerns:
The experimental design is not clear in the methods. Add as many details as possible about the stream reach sampled- which 100 meter stretch (landmark – research station or point of origin or distance from the next tributary it joins). For the sampling protocol, April to August (5 months), every other week (10 possible samples) a 25 meter section of the 100 meters and then the second 25 meter section (samples out of 50 meters)? Then 4 hours during the day and 4 hours at night. So, 2 hours per 25 meters. All spiders collected? So, 10 weeks but 16 quadrats- is this considering quadrats as 50 meter sections? Is the total of 16 samples combined so we have four daytime and 4 nighttime samples for river and forest? This becomes a small sample size and does not reflect seasonal changes in numbers or sizes (which likely impact prey selection). In the results, lines 20-21, the authors report 8 samples (assuming that day and night are combined). Because both spider assemblage and aquatic insect emergence are tightly linked to circadian rhythm it seems that the data should be analyzed independently with time a day as a factor in the analysis (which is true for Tables 1 and 2 as well).
The authors need to clarify Malaise trapping- was this done 16 times also?
Throughout the results/ discussion, the authors should use caution in interpreting their data. For example, page 10, lines 8-9 these groups were most important during this study in this watershed. Without knowing much about the overall insect fauna it is difficult to know if this is selective feeding or the most abundant organisms present.
Table 1 could be presented as means + 1 SE and significance could be tested with a t-test for each taxon
For Table 2, the authors should at least comment in the discussion about the probability that the taxa could be eaten… for example, cicadas are very big and broad-shouldered water bugs are very small and on the water’s surface.
Figure 2 is not very clear and are very similar. It might be better to combine into a single average figure to make more readable.

Minor corrections:
Abstract
nonmetric dimensional scaling (NMDS)
Therefor should be “therefore”
… with their possible prey “was conducted”

Page 2, line 29 ; however,
Page 3, line 6 Theriidae) are also more abundant
Page 3, line 21 insects becoming food
Page 3 Line 25 indent
Page 3 Line 28 10 m from the edge of the riparian shoreline
Page 4 line 19- in 2012 total rainfall was…

Page 6 line 11 were collected at random…
Page 6, line 20 microhabitats were sampled.
Page 6, line 22 … is fixed during the aquatic larval stage.
Page 6, line 27 channel and from a …
Page 7, line 24. Often used only…
Page 7, line 29 each feeding group…
Page 8 line 4 nocturnal samples collected
Page 8 Line 12-15: delete. This sentence repeats info from table
Page 8. Difference in the degree of separation is odd phrasing and unclear.
Page 9, line 14. There was a high degree…
Page 9, line 31 (5.19 0/00)
Page 10, line 17. Emerging aquatic insects have been…
Page 10, line 21 within 1.5m of the stream
Page 14, line 6 found even a few meters…

---

## Round 0.2 · accepted · Accept

Thank you for your response to the reviews and my comments. I believe the manuscript is now ready for publication, and I'm happy to inform you I have accepted it for publication. Upon reading the revised manuscript, I did find some minor stylistic points(mostly capitalizations, use of commas, and a couple word choice issues). These I corrected, and I can supply you a copy of my corrections (although I think it is accessible to you through the PeerJ manuscript page.) A more significant style issue exists with your references, which are in all caps in the revision. I made changes to some of the references but ran out of will power to fix them all. I have queried the PeerJ staff to see whether or not they will fix this or if this is something you will need to do. [NOTE from Staff - our production process will fix this]